# Exosomal Prostate-Specific Membrane Antigen (PSMA) and Caveolin-1 as Potential Biomarkers of Prostate Cancer—Evidence from Serbian Population

**DOI:** 10.3390/ijms25063533

**Published:** 2024-03-21

**Authors:** Suzana Matijašević Joković, Aleksandra Korać, Sanja Kovačević, Ana Djordjević, Lidija Filipović, Zorana Dobrijević, Miloš Brkušanin, Dušanka Savić-Pavićević, Ivan Vuković, Milica Popović, Goran Brajušković

**Affiliations:** 1Centre for Human Molecular Genetics, Faculty of Biology, University of Belgrade, 11000 Belgrade, Serbia; suzana.matijasevic@bio.bg.ac.rs (S.M.J.); milosb@bio.bg.ac.rs (M.B.); duska@bio.bg.ac.rs (D.S.-P.); 2Centre for Electrone Microscopy, Faculty of Biology, University of Belgrade, 11000 Belgrade, Serbia; aleksandra.korac@bio.bg.ac.rs; 3Institute for Biological Research “Siniša Stanković”—National Institute of the Republic of Serbia, Department of Biochemistry, University of Belgrade, 11000 Belgrade, Serbia; sanja.kovacevic@ibiss.bg.ac.rs (S.K.); djordjevica@ibiss.bg.ac.rs (A.D.); 4Innovative Centre, Faculty of Chemistry, University of Belgrade, 11000 Belgrade, Serbia; filipoviclidija29@gmail.com; 5Institute for the Application of Nuclear Energy—INEP, Department for Metabolism, University of Belgrade, 11000 Belgrade, Serbia; zorana.dobrijevic@inep.co.rs; 6Clinic of Urology, Clinical Center of Serbia, Faculty of Medicine, University of Belgrade, 11000 Belgrade, Serbia; ivanvukovic.urolog@gmail.com; 7Faculty of Chemistry, University of Belgrade, 11000 Belgrade, Serbia; la_bioquimica@chem.bg.ac.rs

**Keywords:** prostate cancer, exosomes, PSMA, caveolin-1

## Abstract

Prostate-specific membrane antigen (PSMA) and caveolin-1 are membrane proteins that are overexpressed in prostate cancer (PCa) and are involved in tumor growth and increase in aggressiveness. The aim of the present study is therefore to evaluate PSMA and caveolin-1 proteins from plasma exosomes as effective liquid biopsy biomarkers for PCa. This study included 39 patients with PCa and 33 with benign prostatic hyperplasia (BPH). The shape and size of the exosomes were confirmed by transmission electron microscopy (TEM) and scanning electron microscopy (SEM) analysis. Immunogold analysis showed that PSMA is localized to the membrane of exosomes isolated from the plasma of both groups of participants. The relative protein levels of PSMA and caveolin-1 in the plasma exosomes of PCa and BPH patients were determined by Western blot analysis. The relative level of the analyzed plasma exosomal proteins was compared between PCa and BPH patients and the relevance of the exosomal PSMA and caveoin-1 level to the clinicopathological parameters in PCa was investigated. The analysis performed showed an enrichment of exosomal PSMA in the plasma of PCa patients compared to the exosomes of men with BPH. The level of exosomal caveolin-1 in plasma was significantly higher in PCa patients with high PSA levels, clinical-stage T3 or T4 and in the group of PCa patients with aggressive PCa compared to favorable clinicopathological features or tumor aggressiveness. Plasma exosomes may serve as a suitable object for the identification of potential biomarkers for the early diagnosis and prognosis of PCa as well as carriers of therapeutic agents in precision medicine of PCa treatment.

## 1. Introduction

Prostate cancer (PCa) is the most prevalent cancer diagnosed in men in 2023, accounting for 29% of all cases in men worldwide [1]. According to the data published by the National Institute of Public Health’s “Dr. Milan Jovanovic Batut” in 2021, PCa is the third most commonly diagnosed malignancy and the third leading cause of cancer-related mortality in the Serbian male population [2]. In parallel, an increase in early diagnosed PCa over the last decade has led to better management protocols that are shifting the focus onto discrimination between patients with highly aggressive tumors and those who are eligible for active surveillance [3]. Consequently, there is an urgent need to develop novel diagnostic logarithms and to discover highly specific biomarkers that could be used for monitoring the progression of patients with indolent disease form.

Cancer cells communicate with the tumor microenvironment (TME), composed of cells and the extracellular matrix, by releasing extracellular signals (e.g., exosomes) that alter the landscape of the TME, promoting cancer growth and metastasis in healthy tissue [4]. In addition, recent studies have shown that neuron recruitment and tumor innervation are mediated by the release of exosomes (Exos) [5]. Exos are small extracellular membranous vesicles, 30 to 100 nm in size, derived from multivesicular bodies. They transfer biomolecular cargo between cells, reprograming the pathways of recipient cells [6]. Exos infer the altered content of the original tumor cell in their molecular cargo (e.g., DNA, RNA, proteins, lipids and metabolites), representing a kind of fingerprint of the cellular biochemical status. They are present in all biological fluids (blood, plasma, serum, urine, cerebrospinal fluid, etc.) and, as such, represent a readily available source of biomarkers for non-invasive liquid biopsy procedures [7]. Nevertheless, the diagnostic potential of Exos is hampered by the lack of procedures to enrich the cancer-specific vesicles and vesicular biomarkers.

Prostate-specific membrane antigen (PSMA), known as glutamate carboxypeptidase II, is a transmembrane protein expressed in all types of prostate tissues, especially in carcinoma tissues [8]. PSMA, as a multifunctional protein, is involved in a myriad of processes, such as nutrient uptake and the activation and enhancement of cell survival, proliferation and migration. PSMA receptor function is achieved through dimerization. Overexpression of PSMA in carcinoma tissue, especially in aggressive tumors, could potentially disrupt the growth equilibrium of the prostate gland and increase the metastatic potential of PCa [9]. The ongoing clinical trials for the treatment of advanced PCa relate precisely to the use of PSMA radioisotopes. The FDA recently approved (177Lu)-PSMA-617 for the treatment of PSMA-PET-positive patients with metastatic castrate-resistant prostate cancer (mCRPC), and additional phase III clinical trials are underway [10].

Caveolin-1 is a scaffolding membrane protein involved in the structure of the caveolae and is expressed in all cell types. Caveolin-1 plays an important role in the regulation of cell proliferation, migration and angiogenesis, suggesting a link to cancer development and invasion. Further, caveolin-1 is overexpressed in PCa patients and correlates significantly with the aggressiveness of tumor forms [11].

A recent study by Wang CB et al. has shown that PSMA from the urinary Exos of patients with PCa has a clear diagnostic potential to detect clinically significant PCa [12]. However, the collection of urine to obtain urinary Exos requires a prior digital rectal examination, which can sometimes be uncomfortable for the patient. Therefore, we investigated the potential of plasma exosomal proteins PSMA and caveolin-1 as liquid biopsy biomarkers for PCa in early diagnosis and progression prediction by evaluating their relationships with standard prognostic parameters.

## 2. Results

To check the quality of the Exos isolated with the Total Exosome Precipitation Kit (from plasma) (Thermo Fisher Scientific, Waltham, MA, USA), we first analyzed the morphology and size of the Exos using TEM negative staining. The majority of the vesicles showed a typical cup-shaped morphology with a central depression and were between 30 and 100 nm in size (Figure 1a–c). In addition to TEM, we also examined the quality of the isolated Exos by SEM. In the SEM, the Exos had a round morphology and a uniform size distribution (Figure 2a,b).

No significant differences were found in the concentrations of isolated EVs for representative, randomly selected samples from PCa and BPH patients when using NTA in scattering mode (*p* = 0.16). The mean and median sizes of PCa and BPH Exos were comparable (*p* = 0.78 and *p* = 0.66, respectively) and the size values corresponding to the 10th and 90th percentile demonstrated an insignificant difference (*p* = 0.65 and *p* = 0.83, respectively). The average size distribution curves for EVs isolated from the plasma of PCa and BPH patients are presented in Figure 3.

To clearly establish that the origin of the precipitated plasma Exos is the prostate, the exosome samples of five PCa and five BPH patients were analyzed with the antibody of the PSMA protein, anti-PSMA, by immunogold staining. Figure 4 shows the localization of the exosomal prostate marker PSMA in the plasma Exos. The amount of gold signals between prostate cancer and BPH patients was different (arrows indicate the gold signals). BPH-derived Exos were uniformly stained with immunogold, showing an even distribution of PSMA on the surface, while in PCa-derived Exos, we observed three categories—unlabeled, medium-labeled, and heavily labeled—demonstrating a varying level of PSMA surface distribution.

Western blot analysis detected the typical exosomal marker CD63, as well as prostate-specific PSMA and caveolin-1 in plasma Exos from both samples, PCa and BPH (Figure 5a). Further analysis data demonstrated the enrichment of PSMA in plasma Exos from PCa patients compared to Exos from men with BPH (*p* = 0.043). However, the Western blot quantification results showed no statistical difference in the exosomal caveolin-1 levels between PCa and BPH patients (*p* = 0.349) (Figure 5b). Flow cytometry demonstrated that randomly selected samples (five BPH and five PCa) all express CD9 on their surface (Appendix A).

When the patients with PCa were categorized according to the values of standard prognostic parameters and PCa aggressiveness score, the relative exosomal PSMA protein levels did not differ between the subgroups (Table 1). However, the results obtained indicated that the relative protein levels in plasma exosomal samples were significantly higher in PCa patients with a PSA score > 20 ng/mL than in PCa patients with a PSA score < 20 ng/mL (*p* = 0.033, Figure 6, Table 1). In addition, the relative protein level of exosomal caveolin-1 differed significantly between patients with a T3 or T4 stage and patients with a T2 stage of PCa (*p* = 0.0083, Figure 7, Table 1). Similar results were obtained when the relative protein level of exosomal caveolin-1 was compared in PCa patients with less aggressive and with aggressive PCa (*p* = 0.0061, Figure 7, Table 1).

Correlation tests showed that the relative protein level of PSMA and caveolin-1 in matched exosome samples was not significant for all participants (r = 0.13 and *p* = 0.28). When restricting the correlation analysis to a specific diagnosis, the Pearson correlation coefficients indicate no significant correlation for either PCa patients (r = 0.14 and *p* = 0.4) or BPH patients (r = 0.15 and *p* = 0.42).

## 3. Discussion

Since prostate cancer is generally a slow-growing cancer, its early detection, the differentiation between benign and malignant forms of prostate disease, and the prediction of the aggressiveness of the tumor play a crucial role in the success of treatment and contribute to increasing the 5-year survival rate. Therefore, research efforts in the field have been focused on the search for new, non-invasive or minimally invasive carcinoma-specific biomarkers that would ensure a more appropriate treatment approach and a better quality of life for PCa patients [1].

Until now, tissue biopsy has been regarded as the gold standard for diagnosing PCa. However, the limitation of this method in assessing the heterogeneity of the tumor means that the dynamic progression of the tumor cannot be predicted, ultimately leading to poor quality of life or an increased mortality as a consequence of overdiagnosis and overtreatment. On the other hand, liquid biopsy reflects a more comprehensive picture of the genetic landscape of the cancer, which improves clinical decision making during treatment and patient monitoring [13]. Exos promise enormous potential as bioactive cancer markers, as evidence is accumulating that they are involved in tumorigenesis and tumor progression. This, together with a significant increase in the release of Exos by cancer cells, leads to the conclusion that Exos could be an almost ideal biomarker for cancer screening, diagnosis and prognosis [14]. While our previous research was related to the assessment of internal Exos’ content (non-code RNA) [15], we are now focusing on potential biomarkers on the exosomal membrane. To better understand the functionality and potential biomarker capabilities of Exo, scientists are making enormous efforts to specifically extract cancer-derived Exos [16,17]. It has been previously reported that PSMA expression is associated with a high risk of PCa cancer progression and is overexpressed in PCa patients with marked metastasis [18]. In addition, using two cancer cell lines, Lui et al. [19] confirmed the concept that PCa-derived Exos are highly enriched in PSMA compared to the cell extract. Overall, the aim of the present study was to determine whether exosomal PSMA from the plasma of patients is suitable as a non-invasive biomarker for the diagnosis and prognosis of PCa, and, consequently, as a potential therapeutic agent.

The expression and secretion of caveolin-1 has been shown to be associated with PCa and its progression. Ariotti et al. confirmed the assumption of a conventional release of caveolin-1 within the Exos’ pathway and identified another, non-canonical pathway of caveolin-1 release mediated by the special class of Exos (named C-exosomes) [20].

First, the relative protein concentrations of exosomal PSMA and exosomal caveolin-1 were analyzed by Western blot in exosome samples isolated from plasma in a group of patients with PCa and BPH. The relative protein level of exosomal PSMA was found to be significantly higher in patients with PCa than in BPH patients, while the relative protein level of exosomal caveolin-1 in plasma showed no difference between PCa and BPH patients. Our results are consistent with previously published similar findings regarding the potential role of prostate-specific plasma Exos in differentiating PCa from BPH [21]. It has been suggested that exosomal PSMA in plasma could be used as a valuable biomarker for PCa diagnosis. However, no significant difference was found when PCa patients were segregated based on the values of standard prognostic parameters and cancer progression risk, suggesting that exosomal PSMA from plasma has no prognostic potential in this group of study participants with PCa. Our results therefore partially contradict previous findings on the potential clinical utility of PSMA-positive Exos in assessing tumor aggressiveness, and diagnosing and monitoring metastasis [21,22]. Nevertheless, it should be noted that different criteria were used to assess tumor stage and the risk of progression. In addition, it should be noted that aberrant expression, and loss of PSMA expression have already been demonstrated in patients with mCRPC and in patients with neuroendocrine prostate cancer (NEPC), which could be factors related to the achieved spectrum of PSMA protein levels in a heterogeneous group of PCa patients [23,24].

We compared the relative protein level of exosomal caveolin-1 between groups formed according to the values of prognostic parameters and according to tumor aggressiveness to analyze its potential as a prognostic biomarker. Previously, the expression of caveolin-1 was found to be significantly associated with advanced clinical features of PCa [25,26,27,28]. The results presented in our study, which suggest that the relative protein level of caveolin-1 in plasma-derived Exos is higher in patients with high initial PSA values, in patients with clinical T3 or T4 stages, and in PCa patients with a high risk of cancer progression, are consistent with the presumed oncogenic effect of this protein and previous findings, including those mentioned above.

In the end, no significant correlations were observed in the relative levels of the analyzed exosomal proteins between all patients nor for each patient group individually (PCa and BPH). The possible explanations for the lack of correlation could lie in the different progressive power for the clinical outcomes of PCa patients of PSMA and caveolin-1 in plasma Exos.

## 4. Materials and Methods

### 4.1. Study Participants: Sample and Data Collection

This study was conducted using plasma samples from the Center for Human Molecular Genetics. The participants of this study were diagnosed at the Urology Clinic of the Clinical Center of Serbia. This research was approved by the Ethics Committee of the Clinical Center of Serbia in July 2020 (code 570/8). Written informed consent was obtained from all participants, and this research was conducted in accordance with the Declaration of Helsinki. In the present study, Exos were isolated from the plasma of participants, including 39 patients with PCa and 33 with benign prostatic hyperplasia (BPH).

The study participants were diagnosed according to the standard clinical procedure for the diagnosis of prostate cancer. This clinical procedure includes the performance of the following tests and analyses: measurement of the serum level of prostate-specific antigen (PCA), digital rectal examination (DRE), transrectal ultrasound examination (TRUS), and prostate biopsy with subsequent X-ray examination of the bones. The histopathologic Gleason score was ascertained to determine the histopathologic type of PCa, while the TNM classification was used to evaluate the stage of the cancer. The clinical characteristics and age of the patients were recorded after each participant gave their explicit and voluntary consent.

Patients with PCa were divided into groups based on the values of the standard prognostic parameters: initial serum PSA (PSA ≤ 20 ng/mL; PSA > 20 ng/mL), Gleason score (GS ≤ 6; GS ≥ 7), and clinical stage (T1 and T2; T3 and T4). According to the recommendations of the European Association of Urology (EAU) [29], patients with PCa were divided into the following groups based on the values of the standard prognostic parameters for disease progression: low-to-intermediate-risk group according to the EAU criteria: PCa patients with PSA ≤ 20 ng/mL, GS ≤ 7 and clinical stage T1 and T2; and high-risk group according to the EAU criteria: PCa patients with PSA > 20 ng/mL, GS > 7, or clinical stage T3 and T4. Patients with the presence of distant metastases were assigned to the aggressive group.

### 4.2. Exosome Extraction

Exos were extracted from plasma samples using the Total Exosome Isolation Kit (from plasma) (Thermo Fisher Scientific, Waltham, MA, USA) according to the manufacturer’s instructions. In brief, after a short treatment with proteinase K, the precipitation reagent was added to the plasma samples and the solution was incubated in the refrigerator for 30 min. Then, the solution was centrifuged at 10,000× *g* for 5 min at room temperature. The resulting pellet containing Exos was resuspended in 1× PBS. In addition, Exos were recovered using an original method based on immunoaffinity chromatography with a nanobody-coated matrix that recognizes exosome epitopes in their native conformation. The protocol was previously published and described in detail in Filipović et al.’s work [30]. The relative concentration of Exos was determined by measuring the amount of protein using the Qubit Fluorometer and the Qubit Protein Assay Kit (Thermo Fisher Scientific, Waltham, MA, USA). Their presence in the isolates was confirmed by NTA, transmission electron microscopy, scanning electron microscopy (SEM) and Western blot.

### 4.3. Nanoparticle-Tracking Analysis (NTA)

Nanoparticle-tracking analysis (NTA) was performed using the ZetaView^®^ QUATT instrument (Particle Metrix, Inning am Ammersee, Germany) to determine the size distribution profiles of EVs and to measure the EVs’ concentration. Samples of EVs were diluted up to 1:1000 in 50 mM PBS (sterile 0.1 μm pore syringe filter, Sartorius, Shinagawa City, Tokyo) and all scattering mode measurements were performed at room temperature (RT). The concentration and size ranges were calculated using ZetaView software (Particle Metrix, version 8.05.16 SP3, sensitivity 78%, shutter 100, 11 positions, 2 cycles).

### 4.4. Transmission Electron Microscopy

Transmission electron microscopy (TEM) was used for morphological characterization of the extracted vesicles after negative staining of the samples. The exosome samples were placed on a carbon-coated grid and fixed with 2.5% glutaraldehyde. The Exos were contrasted with 1% phosphotungstic acid and air-dried. Electron micrographs were taken with a Philips CM12 electron microscope (Philips, Eindhoven, The Netherlands) equipped with the SIS MegaView III digital camera (Olympus Soft Imaging Solutions, Münster, Germany).

### 4.5. Scanning Electron Microscopy

The morphological characterization of the eluted Exos was also investigated using scanning electron microscopy (SEM). The fixation of Exos was performed with 2.5% glutaraldehyde in PBS for 10 min at RT. Then, the exosome samples were transferred to metal and air-dried. Prior to imaging, all samples were coated with a thin layer of gold using a sputter coater (Polaron SC503, Fisons Instruments, Ipswich, UK). Electron images were taken with a scanning electron microscope, Tescan FE-SEM Mira 3 XMU (Tescan, Brno, Czech Republic).

### 4.6. Immunogold Detection of PSMA

For this method, a suspension of Exos on coated nickel grids were used. The grids were incubated in blocking buffer (phosphate-buffered saline—PBS containing 1% bovine; serum albumin—BSA and 0.1% Tween 20) for 30 min at room temperature. The grids were then incubated with 100 µL anti-PSMA antibody (1:100, D4S1F, rabbit mAb, #12702, Cell Signaling, Danvers, MA, USA) in blocking buffer for 1 h at 37 °C. After rinsing in five separate drops for 10 min each, the grids were incubated with anti-rabbit IgG conjugated with 10 nm gold particles (ab27241, dilution 1:20, Abcam, Cambridge, UK). After incubation, the sections were rinsed in buffer and distilled water, air-dried, and then analyzed and imaged with a Philips CM12 (Philips/FEI, Eindhoven, The Netherlands) equipped with an SIS MegaView III digital camera (iTEM Olympus Soft Imaging Solutions, Münster, Germany).

### 4.7. Western Blot Analysis

The concentration of proteins in the exosome extracts was determined using the Lawry method [31]. Proteins were prepared for sodium dodecyl sulfate (SDS) polyacrylamide gel electrophoresis by mixing with 2× Laemmle’s buffer (125 mM Tris-HCl, pH 6.8, 4% SDS, 20% (*w*/*v*) glycerol, 10% β-mercaptoethanol, 0.01% bromophenol blue) in a 1:1 ratio and incubating in boiling water for 5 min. Proteins (50 µg) were separated by electrophoresis through 4% stacking gels and 7.5% or 12% separating gels and transferred to polyvinylidene fluoride (PVDF) membranes (Merck Millipore, Burlington, MA, USA). After transfer, the membranes were blocked with bovine serum albumin (3%) for 90 min and incubated overnight at 4 °C with primary antibodies: anti-PSMA (12702) from Cell Signaling, Danvers, MA, USA; anti-caveolin-1 (ab18199) and anti-CD36 (ab59479) from Abcam, Cambridge, UK; and anti-Hsp70 (C92F3A-5) from CiteAb, Bath, UK. Appropriate secondary antibodies, anti-mouse (ab97046) or anti-rabbit (ab6721) conjugated with horseradish peroxidase (Abcam, Cambridge, UK), were applied for 90 min at room temperature. Between the antibodies and after incubation with the secondary antibody, the membranes were extensively washed with Washing Buffer (0.1% Tween-20 in PBS). The immunopositive bands were visualized by the chemiluminescence method using the iBright FL1500 Imaging System and quantified using iBright Analysis Software (Version number 5.2.0) (Thermo Fisher Scientific, Waltham, MA, USA).

### 4.8. Flow Cytometry

Flow cytometry was performed by coating aldehyde/sulphate latex beads (4% *w*/*v*, 4 µm; Sigma-Aldrich, St. Louis, MO, USA) with EVs. Thirty µL beads were coated with 40 µg of total protein from EV-enriched fraction in PBS overnight at 4 °C. The beads were washed 3 times with PBS and blocked for 30 min at room temperature with 200 mM glycine and 30 min with 5% (*w*/*v*) skimmed milk in PBS. The beads were washed 3 times in PBS before adding anti-CD9 antibodies (dilution 1:5) and incubated 1 h at 37 °C. After washing, the beads were analyzed using an FACS Calibur (BD collecting around 1000 events/s). A blue solid state 200 mW laser at 488 nm was used for excitation. The emission was detected at 561 nm (FL2, PE). The positive beads were gated on the FL2-PE, SSC plot. PE-specific fluorescence was assessed as the signal increase with respect to negative control (autofluorescence of beads not coated with EVs, blocked with milk and incubated with antibodies).

### 4.9. Statistical Analysis

The Mann–Whitney U test and the Kruskal–Wallis test were used to determine the statistical significance of PSMA and caveolin-1 protein levels between different sample groups. The Shapiro–Wilk test was applied to assess the normality of the distribution of the results. PSMA protein levels were correlated with caveolin-1 protein levels using Spearman’s test. The results were presented as Pearson’s correlation coefficients (r) and the corresponding *p*-values. Statistical analyses were performed with RStudio 2021.09.1 statistical software. *p*-values < 0.05 were considered statistically significant. Patients with missing data on some of the prognostic parameters were excluded from the association analyses that were relevant for exactly this parameter.

## 5. Conclusions

In this study, we confirmed that the relative protein level of exosomal PSMA in plasma is associated with the development of PCa and could prospectively be used as a potential non-invasive biomarker for PCa. In addition, our study verified that the relative protein level of caveolin-1 from plasma Exos was associated with unfavorable clinical features of PCa and a highly aggressive form of the tumor.

## Figures and Tables

**Figure 1 ijms-25-03533-f001:**
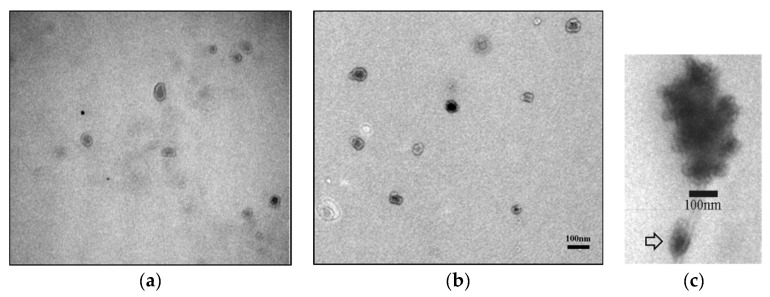
Morphological characterization of Exos by electron microscopy. TEM analysis of a precipitated heterogeneous population of Exos from patients with PCa (**a**) and BPH (**b**) using negative staining showed the typical cup-shaped morphology. Aggregates of precipitated Exos and a single exosome (marked by arrow) (TEM, negative staining) (**c**). TEM—transmission electron microscopy; PCa—prostate cancer; BPH—benign prostatic hyperplasia.

**Figure 2 ijms-25-03533-f002:**
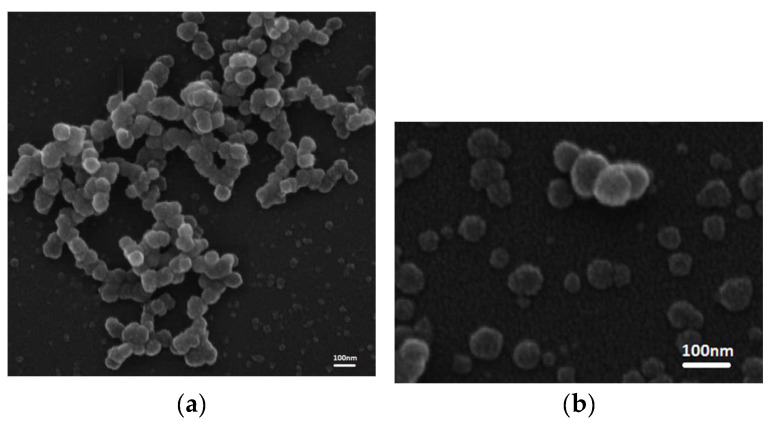
SEM analysis of the precipitated Exos from both group of patients (BPH and PCa, (**a**,**b**), respectively) indicated a round morphology and a uniform size distribution. SEM—scanning electron microscopy; PCa—prostate cancer; BPH—benign prostatic hyperplasia.

**Figure 3 ijms-25-03533-f003:**
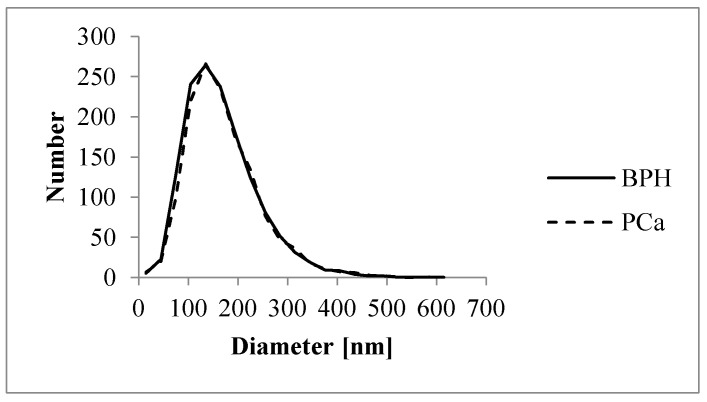
Size distribution of EVs isolated from PCa and BPH samples, shown as a scatter plot. The *X*-axis represents the diameter of detected particles in nm, while the *Y*-axis indicates the absolute number of particles.

**Figure 4 ijms-25-03533-f004:**
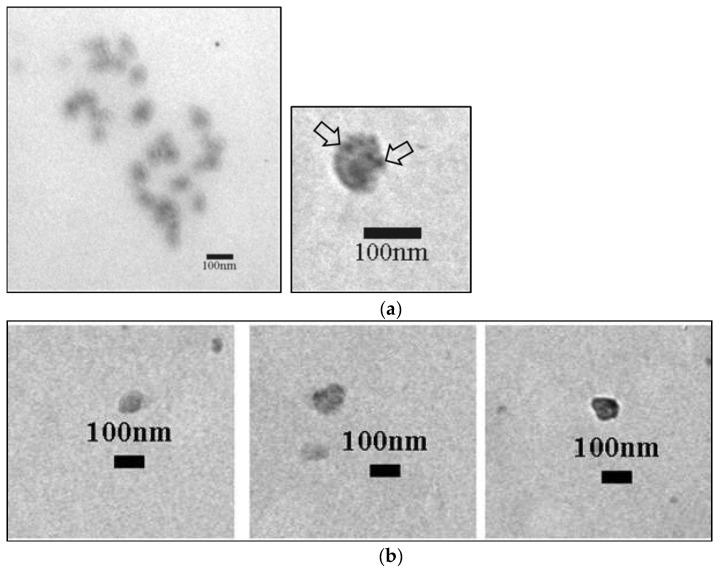
PSMA immunogold labeling of Exos from patients with BPH and patients with PCa (gold signals are marked with arrows). Exos from BPH patients present an even distribution of gold signals (**a**). The images below show different types of labeling of Exos from PCa patients (unlabeled, medium-labeled, and heavily labeled) (**b**). PSMA—prostate-specific membrane antigen; PCa—prostate cancer; BPH—benign prostatic hyperplasia.

**Figure 5 ijms-25-03533-f005:**
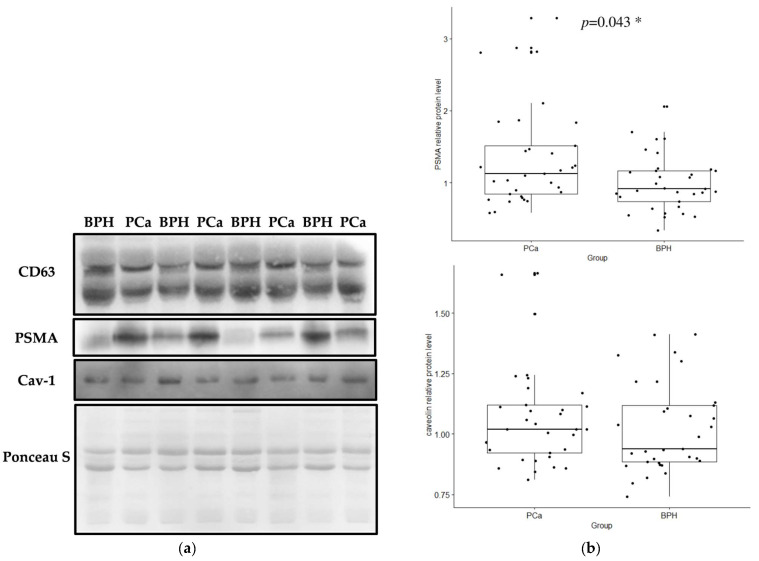
Representative Western blot analysis showing the expression levels of CD63, the common exosome marker, PSMA and caveolin-1 in PCa and BPH samples of plasma-derived Exos and Ponceau S staining as a loading control (**a**). The relative protein levels of PSMA and caveolin-1 in the plasma Exos of PCa and BPH patients were determined by Western blot analysis and normalized to Ponceau S (relative protein level, *y*-axis) (**b**). A Mann–Whitney U test or a Kruskal–Wallis test was used to determine statistical significance. Data are shown as interquartile ranges. The horizontal line in the boxes represents the median values ± SD. Pca—prostate cancer; BPH—benign prostatic hyperplasia; PSMA—prostate-specific membrane antigen, *—statistical significance.

**Figure 6 ijms-25-03533-f006:**
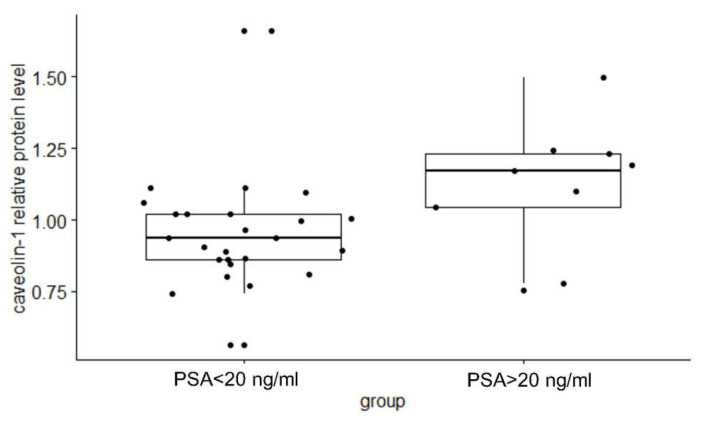
Difference in the exosomal caveolin-1 relative protein level between subgroups of patients with a PSA > 20 ng/mL and patients with a PSA < 20 ng/mL. A Mann–Whitney U-test or a Kruskal–Wallis test was used to determine statistical significance. Data are shown as interquartile ranges. The horizontal line in the boxes represents the median values ± SD. PCa—prostate cancer; BPH—benign prostatic hyperplasia.

**Figure 7 ijms-25-03533-f007:**
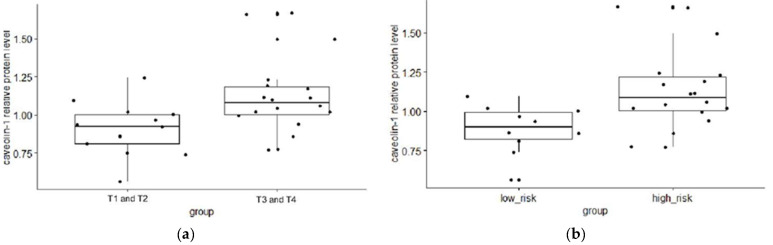
Difference in the exosomal caveolin-1 relative protein level between subgroups of patients with T1 or T2 and T3 or T4 tumor stage (**a**), and patients with low and high risk of cancer progression (**b**). A Mann–Whitney U test or a Kruskal–Wallis test was used to determine statistical significance. Data are shown as interquartile ranges. The horizontal line in the boxes represents the median values ± SD. PCa—prostate cancer; BPH—benign prostatic hyperplasia.

**Table 1 ijms-25-03533-t001:** Differences in the relative proteins level between the patient groups, formed according to the values of the standard prognostic parameters and the aggressiveness of the PCa.

Groups of Patients	Relative Protein Level ± SD	*p* Value
exosomal PSMA		
PSA < 20 ng/mL/PSA > 20 ng/mL	1.05 ± 0.69/1.35 ± 0.85	0.26
T1 and T2 stage/T3 and T4 stage	1.07 ± 0.74/1.24 ± 0.77	0.49
GS < 7/GS ≥ 7	1.15 ± 0.85/1.10 ± 0.70	0.93
low risk/high risk	1.13 ± 0.83/1.29 ± 0.74	0.52
exosomal caveolin-1		
PSA < 20 ng/mL/PSA > 20 ng/mL	0.93 ± 0.20/1.08 ± 0.23	0.033
T1 and T2 stage/T3 and T4 stage	0.89 ± 0.17/1.09 ± 0.26	0.008
GS < 7/GS ≥ 7	1.00 ± 0.32/0.98 ± 0.21	0.51
low risk/high risk	0.87 ± 0.15/1.10 ± 0.26	0.006

## Data Availability

The data supporting the findings of this study are available from the corresponding author upon reasonable request.

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
