# Peer review of "Exosomal Prostate-Specific Membrane Antigen (PSMA) and Caveolin-1 as Potential Biomarkers of Prostate Cancer—Evidence from Serbian Population"

_ijms, 2024, doi:10.3390/ijms25063533_

Round 1
Reviewer 1 Report
Comments and Suggestions for Authors
This is a preliminary clinical study evaluating the level of PSMA in plasma exosomes derived from prostate cancer cells. The findings are consistent with prior research, suggesting that the PSMA-positivity of prostate cancer cells can be monitored using liquid biopsy. However, I have several major concerns that need to be addressed:
-
1-Introduction:
-
-Please reference ongoing clinical trials evaluating this topic.
-
-Compare the advantages and disadvantages of urine-derived versus plasma-derived exosomal PSMA. Refer to and contrast your study focusing on plasma exosomes with the following study on urine-derived exosomal PSMA:
Wang C.-B., Chen S.-H., Zhao L., Jin X., Chen X., Ji J. Urine-derived exosomal PSMA is a promising diagnostic biomarker for the detection of prostate cancer on initial biopsy. Clin. Transl. Oncol. 2023;25:758–767. doi: 10.1007/s12094-022-02983-9.
-
Elaborate on the rationale behind evaluating Caveolin-1 in addition to PSMA. Also, discuss the potential for PSMA-suppression in advanced stages of prostate cancer, for instance:
Neuroendocrine differentiation of prostate cancer leads to PSMA suppression. Endocr Relat Cancer. 2018 doi: 10.1530/ERC-18-0226.
-
-
2-Figures:
- -Figure 5: Please explain why PSMA and CD63 have 8 lanes, while Caveolin-1 and Ponceau S Staining have 9 lanes. Align the samples properly.
- -Figures 6-7: Remove the frame for consistency with the other figures.
- Format: Ensure a consistent format for labeling your figures. Some figures use (A), (B), ..., while others use (a), (b), (c),...
-
3-Results:
-
-The authors report a spectrum of PSMA expression among their samples, which aligns with the emerging understanding of PSMA heterogeneity. In your discussion, please address this point by referring to:
Nat Cancer. 2023 doi: 10.1038/s43018-023-00539-6. Landscape of prostate-specific membrane antigen heterogeneity and regulation in AR-positive and AR-negative metastatic prostate cancer.
-
Additionally, consider providing more details about cancer samples exhibiting lower PSMA levels.
Reviewer 2 Report
Comments and Suggestions for Authors
Title and abstract - no remarks
Introduction - clearly presenting the contemporary literature on the subject and depicting the need for additional biomarkers for PCa, as well as the rationale for the chosen subject for research - PSMA and caveolin-1 in TME exosomes - No remarks
Results - systematically presenting all the way of the authors` research from quality control of EV in the two groups, as well as PSMA and caveolin-1 assessment, nicely visualized on every step - No remarks
Discussion - nicely incorporating authors` results into the contemporary literature, firmly substantiating their conclusion onto the usage of PSMA EV as a potential 'liquid biopsy' for PCa and on caveolin-1 in EV as a prognostic marker in PCa pateints
row 210 - caveoli-1 - typo? Minor
Material and Methods
row 249 - aggressive group? - needs clarifying - Minor
Reviewer 3 Report
Comments and Suggestions for Authors
Dear Authors,
ijms-2876704
Exosomal Prostate Specific Membrane Antigen (PSMA) and Caveolin-1 as Potential Biomarkers of Prostate Cancer– Evidence from Serbian Population by Jokovi et al evaluated the prostate specific membrane antigen (PSMA) and caveolin-1 as the biomarker for PCa, that are present in exosomes. The authors involved study 39 patients and 33 with benign prostatic hyperplasia (BPH). They analyzed the shape and size of the exosomes via by transmission electron microscopy (TEM) and scanning electron microscopy (SEM) analysis. Western blot analysis shows the presence of PSMA in exosomes isolated from the plasma of both groups of participants. The authors concludes that enrichment of exosomal PSMA in the plasma of PCa patients compared to exosomes of men with BPH. The level of exosomal caveolin-1 in plasma was significantly higher in PCa patients with high PSA levels, clinical stage T3 or T4 and in the group of PCa patients with aggressive PCa compared to favorable clinicopathological features or tumor aggressiveness. The authors concludes that plasma exosomes may serve as a suitable object for the identification of potential biomarkers for the early diagnosis and prognosis of PCa as well as carriers of therapeutic agents in precision medicine of PCa treatment. The study is interesting; however, it needs additional data to support these findings.
Major Comments:
1. iThenticate report is very high. The authors need to edit ms and show the originality.
2. The authors can perform LC/MS on these exosomes to identify more biomarkers
3. Does the exosome Cav1 and PSMA expression correlate with tumoral expression of Cav1 and PSMA
4. Are these patients being treated with any drugs during plasma collection.
5. Does any other exosome marker present in these exosomes other than CD63?
Comments on the Quality of English LanguageMinor Edit needed
Round 2
Reviewer 1 Report
Comments and Suggestions for Authors
This version of work has revised and my concerns has been addressed. Therefore, I can recommend this work to be considered for publication.
Reviewer 3 Report
Comments and Suggestions for Authors
Manuscript is partially improved, however the LC/MS study is underway.